# Membrane–Fresnel Diffractive Lenses with High-Optical Quality and High-Thermal Stability

**DOI:** 10.3390/polym14153056

**Published:** 2022-07-28

**Authors:** Xin Liu, Min Li, Bincheng Li, Bin Fan

**Affiliations:** 1Institute of Optics and Electronics, Chinese Academy of Sciences, Chengdu 610209, China; limin414@mails.ucas.ac.cn; 2School of Optoelectronic Science and Engineering, University of Electronic Science and Technology of China, Chengdu 611731, China; 3University of Chinese Academy of Sciences, Beijing 100049, China

**Keywords:** flexible membrane, polyimide, diffractive optics, microstructure fabrication, wave-front error, diffraction efficiency

## Abstract

The membrane–Fresnel diffractive lens (M-FDL) has great potential in the field of high-resolution and lightweight imaging in orbit. However, the M-FDL with high-optical quality and high-thermal stability cannot be fabricated to a standard by the existing processing methods. In this paper, we propose a method for fabricating an M-FDL composed of three steps: the improved repeated spin-coating of the polyimide (PI) membrane, the secondary mucosal method of silica-framed membrane mirror, and the high-precision fabrication of a multi-level microstructure on a flexible, ultrathin membrane substrate. The results show that the root mean square (RMS) of the wave-front error for M-FDL obtained by the above method is 1/28λ (F# = 8.7 at 632.8 nm) with an 80 mm clear aperture, the average diffraction efficiency is more than 70%, the silica-framed membrane mirror possesses approximately 40 times the overall thermal stability of the traditional metal-framed mirror, and the weight is less than 40 g. The measurement results indicate that the M-FDL has high-optical quality and high-thermal stability and can satisfy the imaging requirements.

## 1. Introduction

The realization of high-resolution optical imaging in geostationary Earth orbit (GEO) usually requires a large aperture of the optical system; for example, for 1-m imaging in GEO of great significance, the aperture of the optical system is approximately 20 m [1,2,3]. However, the increase of the aperture of the traditional space telescope has encountered several bottlenecks. The critical problem is the mass of the traditional telescope with a diameter of more than 8 m will exceed the carrying capacity of the current rocket-launch system [2,3,4,5,6]. Additionally, it is difficult to process such a large aperture reflective optical system with existing optical-processing methods. The diffractive-imaging system with the polyimide (PI) membrane material whose thickness is less than 30 μm has the characteristics of being lightweight and having low surface error requirements, space deployable capacity, and easy replication, which has great potential in the field of high-resolution imaging in the geostationary orbit [7,8]. Clearly, the membrane–Fresnel diffractive lens (M-FDL) is the key to the whole diffractive-imaging system. The basic requirements of M-FDL of a diffraction optical-imaging system for space-based lightweight imaging are as follows: the transmission wave-front is good enough to ensure spatial resolution; the diffraction efficiency is high enough to ensure less stray light and high-imaging contrast; the thermal stability of the primary mirror (including diffraction lens and frame) is good enough to ensure space-environment adaptability.

In 2010, an off-axis M-FDL was processed and tested by Lawrence Livermore National Laboratory (LLNL). The PI-membrane substrate was fabricated by NeXolve Corporation, a recognized leader in the research and development of PI membrane material. The number of levels was two, and the diffraction efficiency was only 35% [9]. In 2014, the same company fabricated an M-FDL with a four-level microstructure of 1 um minimum line width by improving the processing technology [10]. The diffraction efficiency and wave-front error of the M-FDL are 55% and 0.63λ at 632.8 nm, respectively, which cannot meet the optical-quality requirements. In 2016, the University of Science and Technology of China used UV–lithography to process diffraction patterns on glass substrate, then reproduced the microstructure patterns on the PI to obtain the M-FDL [11,12]. Because the processing was not mature enough, the M-FDL on the PI membrane that they finally fabricated has only two levels, and the diffraction efficiency was only 33%. In 2020, the Institute of Optics and Electronics, Chinese Academy of Sciences, proposed an aerated-membrane method for directly fabricating M-FDL [13,14]. The wave-front error of the M-FDL they finally fabricated can reach 0.035λ at 632.8 nm. However, this method cannot achieve the processing of a multi-level microstructure due to the limitation of the technical principle; thus, the diffraction efficiency is only 38%.

In the above studies, the problem of improving the diffraction efficiency and wave-front accuracy of the M-FDL has not been solved well. The key factor is that the multi-level microstructure on a flexible, ultrathin PI-membrane substrate cannot be fabricated with high precision (the position accuracy and 3D-morphology accuracy of the microstructure) by existing processing methods. In addition, the current frames are all metal materials, which have poor dimensional stability in a space environment. As a result, the imaging quality of a diffraction optical-imaging system cannot be guaranteed. Thus far, an M-FDL with high-optical quality (high-diffraction efficiency and low-transmission wave-front errors) and high-thermal stability for imaging applications has not been reported. Therefore, it is urgent to develop a high-precision fabrication method for M-FDL with the above characteristics.

In this paper, a method of fabricating M-FDL with high-optical quality and high-thermal stability for imaging applications is presented. Firstly, the PI membrane is fabricated by improved repeated spin-coating processes. The PI membrane is tensioned and fixed by annular fixtures. Then, a ‘secondary mucosal method’ is used to obtain a silica-framed membrane mirror with great thermal stability. Finally, we propose a high-precision fabrication method for preparing a multi-level microstructure on a flexible, ultrathin PI-membrane substrate. By the above method, we successfully fabricate an M-FDL with a diameter of 110 mm and a weight of less than 40 g. The silica-framed membrane mirror possessed approximately 40 times the overall thermal stability of the traditional metal-framed mirror. The wave-front error of the M-FDL was approximately 1/28λ at 632.8 nm, and the average diffraction efficiency was more than 70%. The experiment results indicate that the M-FDL has high-optical quality and high-thermal stability and can satisfy the imaging requirements. Furthermore, the M-FDL can be mass fabricated using our method, which will promote its extensive application in scientific research and engineering practices.

## 2. Imaging Theories of M-FDL

Considered from the perspective of imaging functions, the traditional lens is based on refraction and reflection principles, and uses a refractive lens, prisms, reflective lens, and other conventional optical devices to focus with each ray emitted from an object point having the same path to the image point [15]. The basic structure of a diffractive lens is illustrated in Figure 1a. By removing the multiple 2π phase delays from the refractive lens, one obtains a harmonic diffractive lens (HDL), and one can think of a diffractive lens as a modulo 2π lens [16]. The blaze profile within each zone of the diffractive lens provides a perfect constructive interference at the focal plane for the design wavelength, *λ*_0_ (the wavelength that experiences a 2π phase jump at each zone boundary).

In theory, the diffractive efficiency of an HDL with three-dimensional (3D) continuous morphology can be increased to 100%. However, for the existing micro/nano-fabrication technology, it is very difficult to manufacture a continuous 3D-morphology microstructure on the flexible substrate. Therefore, the Fresnel diffractive lens (FDL) with a multi-level microstructure is often used to replace HDL with continuous surfaces in the actual processing. Schematic diagrams of two-level and four-level microstructures are shown in Figure 1b.

In theory, the thickness of the diffractive lens needs only be greater than the depth of the microstructure providing the phase modulation function, which can be calculated by Equation (1).
(1)$$h=\frac{\lambda_{0}}{n_{0}-1}$$

As concerns the FDL, the thickness of the substrate is much greater than this value; thus, the weight of the FDL is almost equal to the weight of the substrate. To ensure surface figure precision, traditional diffractive lenses need a certain radius-to-thickness ratio; hence, they are large in volume and heavy in weight [17]. Because the thickness of the PI substrate used in the M-FDL is less than 30 μm, and the surface density of the primary mirror is lower than 1 kg/m^2^, which contributes to the lightness of the optical system.

Another fundamental difference is that the traditional refractive lens has only one focal point, but the FDL has many focal points, as shown in Figure 1c. When the incident wavelength is *λ*, the focal length, *f**_λ_*, can be calculated by Equation (2).
(2)$$f_{\lambda }=\frac{\lambda_{0}\cdot f_{0}}{i\cdot \lambda }$$
where $f_{0}$ is the main focal length at a given wavelength, *λ*_0_, and *i* is the diffractive order. Thus, FDL has a higher degree of freedom than a traditional lens in terms of optical design, material selection, and dispersion characteristics [18]. In addition, the FDL is a transmission optical element, which can greatly reduce the requirement for surface tolerance.

As the level number increases, the diffraction efficiency is greatly improved, as shown in Figure 1d. The diffraction efficiency can be calculated by Equation (3).
(3)$$\eta =\frac{1}{L^{2}}\cdot \sin c^{2}\left(\frac{i}{L}\right)\cdot \frac{\sin^{2}\left[\pi \left(\alpha -i\right)\right]}{\sin^{2}\left[\frac{\pi \left(\alpha -i\right)}{L}\right]}$$
where *L* is the number of levels of the microstructure, and $\alpha =\lambda_{0}/\lambda$ is the wavelength-matching factor. When the number of levels is two, the theoretical maximum diffraction efficiency is only 40.5%. When the number of levels is four, the theoretical maximum diffraction efficiency increases to 81.0%. In this paper, the four-level M-FDL with a higher diffraction efficiency is our object of study.

## 3. Fabrication of M-FDL

### 3.1. Preparation of the Ultrathin Flexible PI Membrane

From the perspective of aerospace environment applications, scientists chose PI as the ultrathin flexible membrane substrate material. The coefficient of thermal expansion (CTE) of the PI material is near zero [19,20,21].

In this paper, we use the spin-coating technology to prepare the PI film. In semiconductor processes, the spin-coating method is usually used to coat photo-resistant film with a viscosity between 1 mPa and 2000 mPa. The peak value (PV) of the thickness uniformity error of the film is approximately 5%. For PI film with thickness up to 25 μm, the PV error reaches 1.25 μm, which is approximately 2λ at 632.8 nm. It cannot meet the optical quality requirements of the PI substrate (PV < 0.2λ, RMS < 0.02λ at 632.8 nm).

Therefore, we improve the spin-coating method in three aspects, as shown in Figure 2a. Firstly, the method of one-step spin-coating a thick solution is changed to multi-step spin-coating a thin solution. This could reduce the thickness-uniformity error of the PI film caused by the thick-edge effect (edge bead). Secondly, before each spin-coating, the spin-coater is adjusted with an electronic level and adjustment machine to ensure the silica substrate level. Thirdly, after each spin-coating, the silica substrate is placed on a hot plate with good flatness and levelness where the precursor PAA solution on the silica is pre-cured.

During the spin-coating, the flow of the solution on the silica substrate is mainly affected by three factors: the centrifugal force generated by the rotation, the viscosity of the solution, and the evaporation rate of the solvent in the solution [22,23]. The evaporation rate of the solvent is not a constant and is influenced by the rotational speed, the concentration of the solution, and the nearby exhaust flow. When the other parameters are fixed, the relationship between film thickness and several key parameters is shown in Figure 2b.

Based on the above research, we prepared an ultrathin flexible PI membrane, as shown in Figure 2c. In addition, the performance parameters of the PI materials are measured. It is worth mentioning that the coefficient of thermal expansion is measured by a thermal-mechanical analyzer (TMA) and the specific heat capacity is measured by differential-scanning calorimetry (DSC). The measurement results are shown in Table 1, which indicate that the PI membrane we prepared has high-thermal stability and excellent mechanical properties.

### 3.2. Fixation of the Ultrathin Flexible PI Membrane with High-Thermal Stability

To ensure the excellent optical performance and environmental adaptability of the M-FDL, the ultrathin flexible PI membrane substrate must have high flatness, great mechanical properties, and thermal stability. Therefore, the PI membrane in the free state cannot be used directly and must be tensioned with fixed devices.

In this paper, to obtain uniform radial pre-stress, as shown in Figure 3a, two annular fixtures with certain fitting tolerances are first used to tighten and fix the ultrathin flexible PI membrane. The whole fixtures are made of metal aluminum (2A12), created through mechanical processing, and the flatness of the contact surface between the fixtures and the PI membrane is better than 10 μm. Next, the two fixtures are tightened with screws arranged at a circumferential interval of 30 degrees. The radial pre-stress is determined by the fit tolerances of the two fixtures and screws. Finally, a metal-framed membrane mirror with excellent flatness can be obtained, as shown in Figure 3b. The uniform radial pre-stress is 7.9 MPa, calculated by using a designed experiment and the modified Von Karman’s equation [24].

However, there is a problem when the metal-framed membrane mirror is applied in a space environment. When the temperature varies widely, the metal-framed membrane mirror has poor dimensional stability. As a result, the imaging quality of the diffraction-optical-imaging system cannot be guaranteed.

Therefore, we further propose the ‘secondary mucosal method’ of the flexible PI-membrane substrate and the schematic diagram of the principle is shown in Figure 3c. Firstly, one side of the fused silica ring is uniformly coated with UV adhesive NOA61 and horizontally placed above the membrane. Then, the adhesive is completely cured so that the membrane with good flatness is bonded to the silica ring. Finally, the membrane is split along the outer edge of the silica ring to obtain a silica-framed membrane mirror (surface without diffraction microstructure), as shown in Figure 3d. As the CTE of silica ring is only approximately one fortieth that of the aluminum ring, the overall thermal stability of the membrane mirror prepared by this method can be greatly enhanced.

### 3.3. High-Precision Fabrication of M-FDL with High Optical Quality

To obtain an M-FDL with high-optical quality, we propose a high-precision fabrication method to prepare a multi-level microstructure on flexible ultrathin PI-membrane substrate.

The technical process of the fabrication method is shown in Figure 4.

Firstly, a silica-framed membrane mirror (a surface without a diffraction microstructure) and a supporting plate glass are prepared, as shown in Figure 4a. The upper surface of the supporting plate glass has good flatness. Then, the silica-framed membrane mirror is placed horizontally and coaxially on the glass. Air is exhausted through the vents, as shown in Figure 4b. The negative pressure binds the membrane tightly to the upper surface of the plate glass. The above operations enable the flexible membrane to be directly processed with a high-precision micro/nanostructure, similar to the common hard substrate.

Next, the photo-resistant material is spin-coated and soft baked on the membrane substrate, as shown in Figure 4c. After that, we transfer the mask patterns to the photo-resistant material using contact-printing technology and wet-development technology, as shown in Figure 4d,e. Finally, the two-level microstructure is fabricated on the PI-membrane substrate by reaction ion etching (RIE)-technology. Figure 4g shows the two-level Fresnel diffractive microstructure on the ultrathin flexible PI-membrane substrate.

By repeated steps and multiple overlay exposures, as described in Figure 4k, a four-level M-FDL with higher diffraction efficiency can be obtained, as shown in Figure 4n. Significantly, to ensure that the air between the membrane and plate glass can be exhausted completely and to prevent plastic deformation of the PI membrane above the exhaust vents, the negative pressure is approximately 0.2–0.4 bar, and the diameter of the exhaust vents should not be larger than 1 mm.

According to the technical process, the experiment is carried out. In this paper, the inner and outer diameters of the silica ring are 90 and 110 mm, respectively. Therefore, the clear aperture is designed to be 80 mm. Some key optical parameters of the M-FDL and the dimension parameters of the diffractive microstructure are shown in Table 1.

By the ‘secondary mucosal method’ we propose, the PI membrane is fixed with a silica ring, the CTE of which is only 0.64 × 10^−6^/°C. The silica-framed membrane mirror is placed horizontally and coaxially on the plate glass. Then, air is exhausted through the vents at 0.2 bar to ensure that the flexible membrane is closely attached to the plate glass and becomes a solid substrate.

Next, the uniformly positive photo-resistant AZ1500 is spin-coated at 2000 revolutions per minute (rpm) on the surface of the PI membrane. Then, the substrate with a platen glass is placed under the membrane on a hot plate at 100 °C for 90 s. After cooling at room temperature, the membrane substrate with a photo-resistant surface is exposed by a contact aligner at an exposure dose of 40 mj/cm^2^. The exposed photo-resistant surface could be removed by the developer AZ300MIF (TMAH, 2.38%). By rinsing with DI water and post-baking in an oven at 120 °C for 5 min, a two-level microstructure on photo-resistant material is obtained.

Finally, we transfer the microstructure to the surface of the PI membrane by RIE technology. The etching rate is 45 nm/min when the gas flow of O_2_ is 180 sccm, the radio frequency power is 500 W, and the temperature of the cooling water is 15 °C. By repeated steps and multiple overlay exposures, a four-level M-FDL with higher diffraction efficiency can be obtained, as shown in Figure 5.

## 4. Results and Discussion

The M-FDL with high-optical quality and high-thermal stability is shown above. The diffractive effect is obvious to human eyes. Encouragingly, the total weight of the M-FDL is only 39.8 g, approximately one-tenth that of a traditional refracting lens with the same aperture.

We use the OLYMPUS-STM7 optical microscope to observe the surface of the M-FDL, the results of which are shown in Figure 6a. We can see clear boundaries between the different bands, and a period can be seen every four bands. The surface of the microstructure is smooth and without defects.

The geometric-morphology parameters of the M-FDL are measured using a Dektak-XT surface profilometer, and the results are shown in Figure 6b. The microstructure height of each level is basically the same, indicating that the morphology is sufficient. However, the microstructure has a width error of not more than 0.5 μm due to alignment errors during the overlay exposure process, which is inevitable in principle.

We built an optical detection path to test the transmission wave-front of the M-FDL, including an interferometer, a lens with F# = 7, an adjusting frame with six degrees of freedom, and a standard plane mirror. The measurement result is shown in Figure 6c. The root mean square (RMS) of the wave-front error is 22.7 nm, which is approximately 1/28λ at 632.8 nm (<1/20λ). This means that the fabricated M-FDL completely meets the requirements for ideal optical imaging.

In addition, the measurement results of diffraction efficiency are shown in Figure 6d. The average diffraction efficiency is 71.4%, close to the value of 88% of the theoretical maximum diffraction efficiency, satisfying the use requirements of the M-FDL. The above measurement results prove the feasibility of the M-FDL for imaging applications.

## 5. Conclusions

Due to the disadvantages of the traditional fabrication approaches, a fabricating method, M-FDL, with high-optical quality and high-thermal stability for imaging applications is proposed in this paper. A ‘secondary mucosal method’ is used to obtain a silica-framed membrane mirror with great-thermal stability, and then, the multi-level microstructure on the flexible ultrathin PI-membrane substrate is prepared by our proposed high-precision fabrication method. The experiment results show that the transmission wave-front of the M-FDL is approximately 1/28λ at 632.8 nm, and the average diffraction efficiency is more than 70%, meeting the imaging requirements of the optical systems. In general, our method may help promote the extensive applications of Fresnel diffractive lenses on ultrathin flexible membranes in scientific research and production.

## Figures and Tables

**Figure 1 polymers-14-03056-f001:**
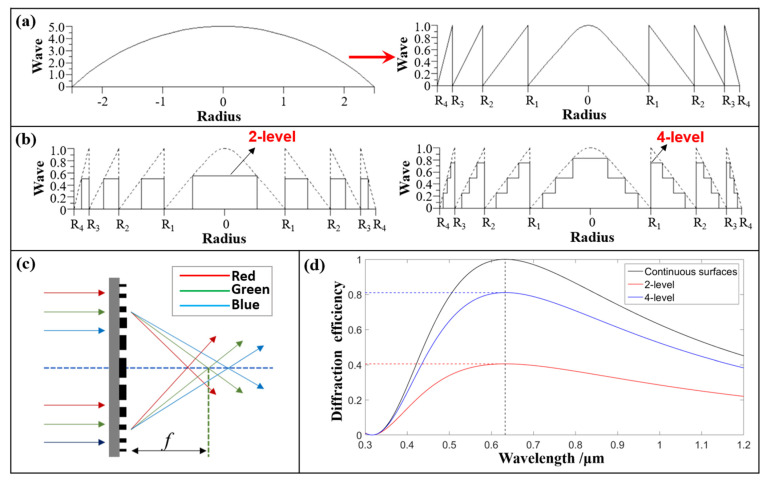
Imaging principle and optical property of FDL. (**a**) Origin and evolution of a diffractive lens. (**b**) Schematic diagrams of 2-level and 4-level microstructures. (**c**) Dispersion property. (**d**) Diffractive efficiency property.

**Figure 2 polymers-14-03056-f002:**
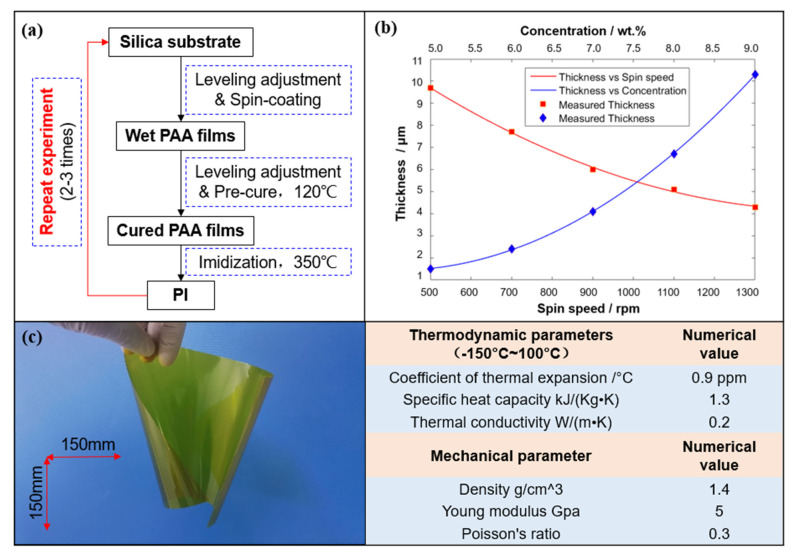
Preparation of the PI membrane. (**a**) Process of repeated spin-coating. (**b**) Membrane-thickness variation with different parameters: spin speed and solute concentration and fluid viscosity. (**c**) Image and characteristic parameters of PI membrane in the free state.

**Figure 3 polymers-14-03056-f003:**
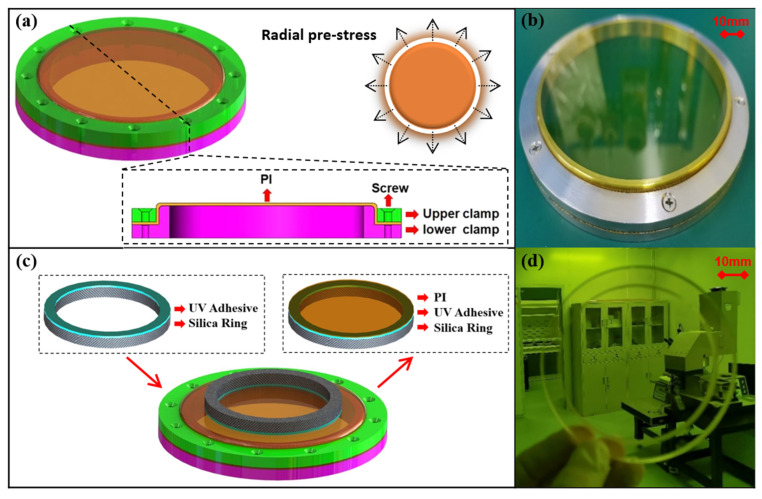
Fixation method of the PI membrane. (**a**) Structure of the metal-framed membrane mirror. (**b**) Image of the metal-framed membrane mirror. (**c**) Structure of the silica-framed membrane mirror. (**d**) Image of the silica-framed membrane mirror.

**Figure 4 polymers-14-03056-f004:**
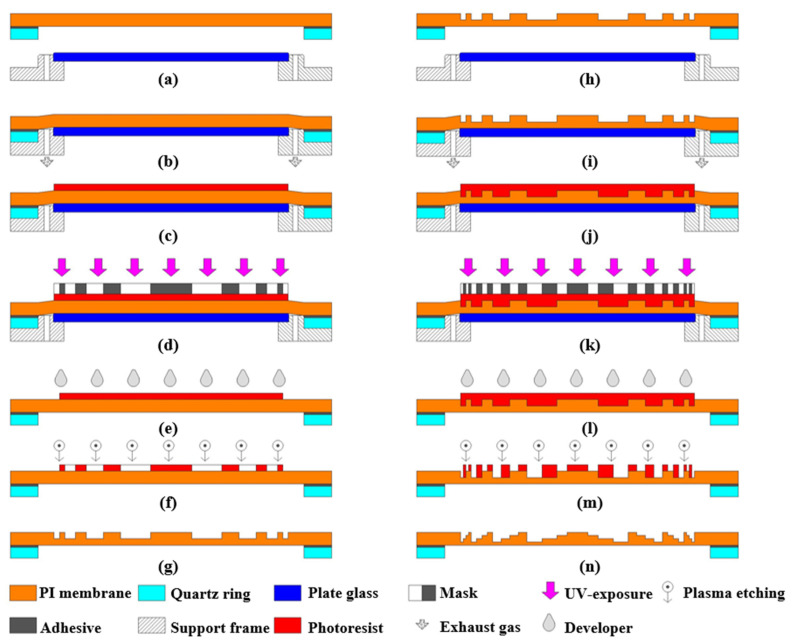
Schematic of the technical process. (**a**) The silica-framed membrane mirror and supporting plate glass. (**b**) The membrane is attached to flat glass with no gap. (**c**) First spin-coating. (**d**) Contact exposure. (**e**) First development. (**f**) First reactive ion etching. (**g**) Two-level Fresnel diffractive microstructure. (**h**) The silica-framed membrane mirror and supporting plate glass. (**i**) The membrane is attached to flat glass with no gap. (**j**) Second spin-coating. (**k**) Overlay exposure. (**l**) Second development. (**m**) Second reactive ion etching. (**n**) Four-level Fresnel diffractive microstructure.

**Figure 5 polymers-14-03056-f005:**
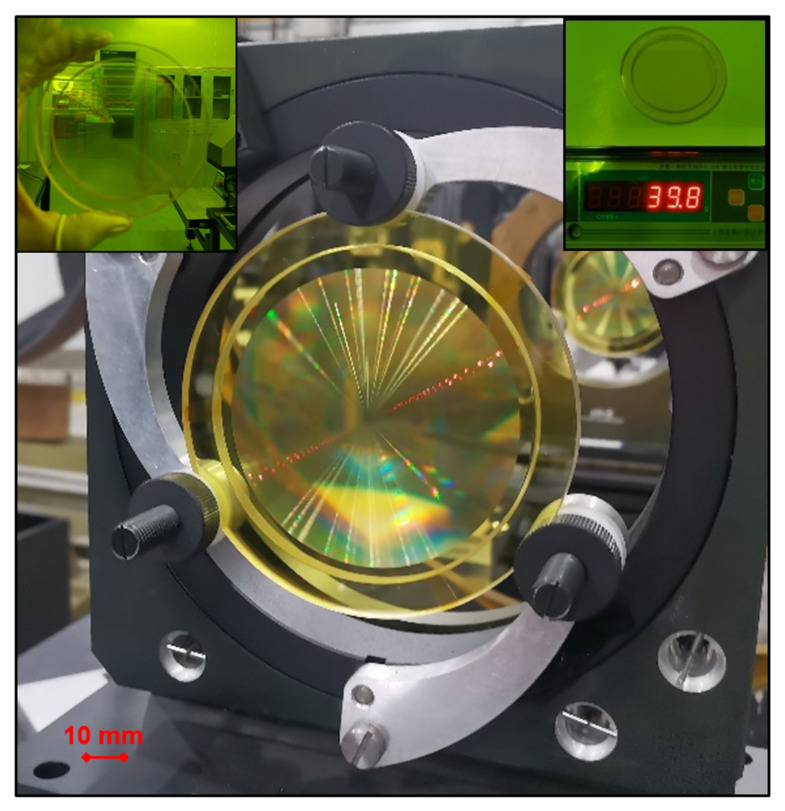
The membrane–Fresnel diffractive lens (M-FDL).

**Figure 6 polymers-14-03056-f006:**
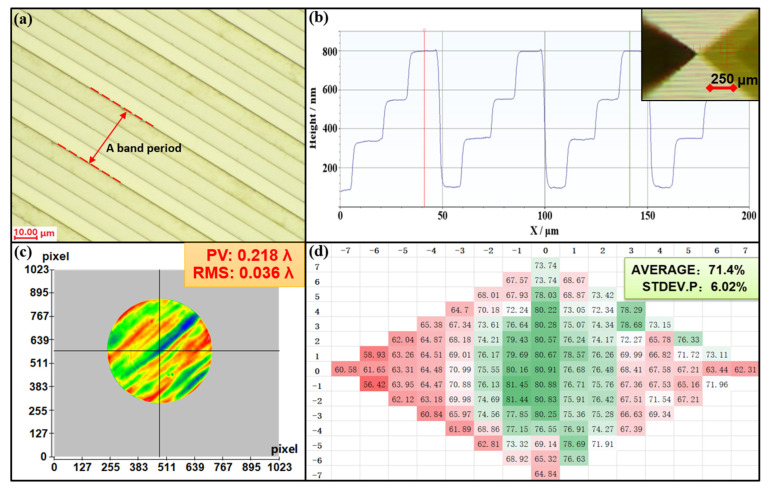
The measurement results of the M-FDL. (**a**) Microscope image. (**b**) The surface profile of the microstructure. (**c**) The measurement results of the transmission wave-front. (**d**) The measurement results of the diffraction efficiency.

**Table 1 polymers-14-03056-t001:** Optical and dimension parameters.

Optical Parameters		Microstructure Parameters	
Central wavelength	632.8 nm	Band number	1819
Focal length	695 mm	Critical dimension	2.75 μm
F_number	8.7	Level height	240 nm

## Data Availability

Not applicable.

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
