# Peer review of "Membrane–Fresnel Diffractive Lenses with High-Optical Quality and High-Thermal Stability"

_polymers, 2022, doi:10.3390/polym14153056_

Round 1
Reviewer 1 Report
This paper demonstrates high-optical-performance Fresnel diffractive lenses by using polymer-based membranes. The authors applied conventional photolithography process with special polymer-based membranes to fabricate the outstanding Fresnel diffractive lenses. Thus, this manuscript showed significant results for the broad researchers. However, the authors should resolve major revisions for further consideration for the publication in Polymers as below:
1. First of all, I should point out the inconsistency of the denotation and the authorship. In the main denotation of the authors, although all the authors are the corresponding authors, the below indication showed just one corresponding author. The authors need to double-check this part.
2. The readability is too weak due to the small size of the words in Figure 1, Figure 2b, and Figure 6c and 6d. I recommend that the word size should increase more.
3. The author needs to add the scale bars in the figures showing camera images and microscope images especially in Figure 2c, Figure 3b and 3d, Figure 5, and inset of Figure 6b.
4. The order of the contents in this manuscript is quite crowded. I recommend that the authors can separate the sections for main body as follows:
A. 2. Theory of the imaging for M-FDL
B. 3. Fabrication of one-step FDL and M-FDL
C. 4. Results and discussions
D. 5. Conclusion
5. For the proper depth of the lenses, I recommend adding more details of the fabrication process. Especially, authors need to indicate what kinds of gases and power used and its etching rate.
6. After the vacuuming between the PI-based membrane and the holder, is it strong enough to handle the membrane during the photolithography? And if yes, what was the principle for the strong strength?
7. For the thermal stability, although the authors mainly insisted that the membrane-based lenses can have high thermal stability by using the silica, however, I was not able to observe the clear evidence in the experimental data. In particular, there was no direct measurement of the thermal stability of the membrane. If authors would like to add the performance in the main contents, I would like to suggest that the authors should add more experimental data or add some references at least.
8. I would like to recommend removing words “novel” used in this manuscript because this word is too huge and unnecessary to appeal. The reader can understand the important points in this work even without the words.
9. Finally, the manuscript should be conducted English editing and dedicated proof-reading.

Reviewer 2 Report
see attached file

Round 2
Reviewer 1 Report
The authors have completed the revision, and I recommend this manuscript to be published.